# Hollow mesoporous atomically dispersed metal-nitrogen-carbon catalysts with enhanced diffusion for catalysis involving larger molecules

Xu Han[1,2], Tianyu Zhang [1,2], Xinhe Wang[1], Zedong Zhang[1], Yaping Li[1], Yongji Qin[1], Bingqing Wang[1], Aijuan Han [1✉] & Junfeng Liu [1✉]

Single-atom catalysts (SACs) show great promise in various applications due to their maximal atom utilization efficiency. However, the controlled synthesis of SACs with appropriate porous structures remains a challenge that must be overcome to address the diffusion issues in catalysis. Resolving these diffusion issues has become increasingly important because the intrinsic activity of the catalysts is dramatically improved by spatially isolated single-atom sites. Herein, we develop a facile topo-conversion strategy for fabricating hollow mesoporous metal-nitrogen-carbon SACs with enhanced diffusion for catalysis. Several hollow mesoporous metal-nitrogen-carbon SACs, including Co, Ni, Mn and Cu, are successfully fabricated by this strategy. Taking hollow mesoporous cobalt-nitrogen-carbon SACs as a proof-of-concept, diffusion and kinetic experiments demonstrate the enhanced diffusion of hollow mesoporous structures compared to the solid ones, which alleviates the bottleneck of poor mass transport in catalysis, especially involving larger molecules. Impressively, the combination of superior intrinsic activity from Co-N$_4$ sites and the enhanced diffusion from the hollow mesoporous nanoarchitecture significantly improves the catalytic performance of the oxidative coupling of aniline and its derivatives.

[1] State Key Laboratory of Chemical Resource Engineering, Beijing University of Chemical Technology, Beijing 100029, China. [2]These authors contributed equally: Xu Han, Tianyu Zhang. ✉email: hanaijuan@mail.buct.edu.cn; ljf@mail.buct.edu.cn

Since the concept of single-atom catalysts (SACs) was developed by Zhang, Li and Liu et al.[1], SACs have received tremendous attention in the past decade due to their extraordinary catalytic activity and exclusive selectivity resulting from atomically dispersed metal sites[2–5]. To date, many strategies for synthesizing SACs have been developed, among which direct pyrolysis of zinc-metal bimetallic zeolite imidazolate frameworks (ZnM-BZIFs) is a prevailing approach to preparing SACs featuring atomically dispersed metal-nitrogen-carbon (MNC) structures[6–9]. The target metal atoms separated by the organic linkers and Zn atoms in the zeolite imidazolate frameworks (ZIFs) atomically dispersed on the derived nitrogen-carbon (NC) during the follow-up pyrolysis with the evaporation of Zn atoms[10,11]. Benefitting from the ordered porous structures of ZIF precursors, the resulting MNC-type SACs usually possess abundant narrow micro/mesoporous structures and a high specific surface area, leading to extraordinary catalytic performance in catalytic reactions involving small molecules such as oxygen, nitrogen, and carbon dioxide[12–16]. However, their applications in organic catalysis, especially in reactions, such as coupling reactions, that involve larger molecular reactants/products, are not yet fully utilized for the following reasons: (1) the narrow micro/mesoporous structures of MNC limit the internal diffusion of large molecules within the MNC and decrease the accessibility of the inner active sites; (2) steric hindrance may also suppress the formation of the final products[17–23].

Hollow structured nanomaterials have been widely studied as efficient catalysts for improving catalytic properties[24–29]. Constructing hollow mesoporous structured SACs containing large enclosed cavities and thin shells with mesopores would be an effective route to increasing the accessible active sites, boosting diffusion, and reducing the effect of steric hindrance. The hollow nanoarchitecture enables the inner and exterior spaces to be easily accessible for reactant molecules and would enrich the reactants as molecular reactors[30–32]; the existing mesopores in thin shells as the molecular channel significantly enhance the diffusion of reactants and products. For example, Li's group reported the preparation of single atomic sites anchored on hollow N-doped carbon spheres as a highly efficient oxygen reduction reaction catalyst[33]. Jiang's group prepared single Zn atoms anchored on hollow N-doped carbon for efficient $CO_2$ conversion[34]. Despite the merit of increased numbers of accessible active sites and short diffusion pathways, currently reported hollow SACs mainly focus on creating void space via the hard template method; however, the limitations on the diffusion of large molecules from the mesopores and narrow channels in the shell are not well settled. The construction of hollow SACs with well-distributed mesopores on the shell is highly desirable. Recently, hollow metal-organic frameworks (MOFs) with hierarchical pores, usually including mesopores on the thin shell and macropores in the center void, have demonstrated great promise for size/shape-selective catalysis[35–37]. Inspired by those studies, we realized that a well-designed MOF with hierarchical pores would be an ideal precursor to the desired SACs to solve the diffusion problem in the catalysis of large molecules.

Herein, we report the design and preparation of hollow, mesoporous, atomically dispersed metal-nitrogen-carbon (h-MNC) catalysts via a topo-conversion strategy by using well-designed MOFs with hierarchical pores as precursors to remit the diffusion limitation of catalysis involving larger molecules. Rational chemical etching of bimetallic ZIFs (BZIFs) produced hollow mesoporous BZIFs, which topologically transformed into h-MNCs after pyrolysis. Aberration corrected high-angle annular dark-field scanning transmission electron microscopy (AC-HAADF-STEM) and X-ray absorption fine structure (XAFS) of the synthesized h-MNCs confirmed the atomic dispersion of

metal atoms. These h-MNCs feature unique hierarchically porous structures with enlarged mesopores of ~10 nm in the shell and hollow cavities of ~200 nm within the h-MNCs; these pores maximize the intraparticle diffusion rate of reactants and products, especially large molecules, and thus enhance the accessibility to active sites and the high utilization efficiency of catalytic active sites. In an impressive combination of the advantages of both the atomically dispersed active sites and hierarchically porous structures, the proof-of-concept hollow, mesoporous, atomically dispersed CoNC (h-CoNC) exhibited much higher catalytic activity with a turnover frequency (TOF) of 586 h$^{-1}$ in the oxidative coupling of aniline into azobenzene at room temperature, surpassing the hollow counterpart with Co nanoparticles (h-Co$_{NP}$NC), the atomically dispersed metal counterpart on solid NC (s-CoNC), and most of the reported state-of-the-art catalysts. In addition, this strategy is quite versatile and has been extended to the synthesis of various h-MNCs with different metal compositions, including Ni, Mn, and Cu, creating more opportunities for optimizing catalysts with high performance.

## Results and discussion

**Morphological characterization of h-CoNC.** The h-MNC catalysts were prepared by a topo-conversion strategy using the well-designed hollow mesoporous MOF as precursors, as schematically illustrated in Fig. 1a. ZnM-BZIFs (M=Co, Ni, Mn, Cu) were synthesized with a mixture of metal acetates and 2-methylimidazole (2-MIM) in a cetyltrimethyl ammonium bromide (CTAB) aqueous solution. The ZnM-BZIFs were then chemically etched with tannic acid (TA) to form hollow mesoporous ZnM-BZIFs. Finally, the desired h-MNCs were fabricated by direct pyrolysis of hollow mesoporous ZnM-BZIFs at 900 °C for 2 h in Ar. This strategy was first demonstrated by using h-CoNC as a proof-of-concept, which was synthesized with zinc acetate and cobalt acetate at a molar ratio of 60:1 as precursors. As shown in the transmission electron microscope (TEM) image in Supplementary Fig. 1, ZnCo-BZIF showed a well-defined truncated rhombic dodecahedral shape with an average size of ~260 nm, which was well retained in the hollow mesoporous ZnCo-BZIF (h-BZIF) sample that had a hollow cavity at the center, and a large number of mesopores on the shell appeared (Fig. 1b and Supplementary Fig. 2). After pyrolysis, the obtained h-CoNC preserved the hollow truncated rhombic dodecahedral shape with a shell thickness of ~30 nm and a void cavity of 200 nm, and no obvious metal nanoparticles were observed on the shells (Fig. 1c). Only discrete bright dots were observed on the AC-HAADF-STEM image, demonstrating the atomic dispersion of the Co atoms. Energy-dispersive X-ray spectroscopy (EDX) mapping of h-CoNC suggested that C, N, and Co were uniformly distributed on the hollow shells (Fig. 1e). The content of Co in h-CoNC was ~0.42 wt%, as determined by inductively coupled plasma-optical emission spectrometry (ICP–OES) (Supplementary Table 1). The X-ray diffraction (XRD) patterns of h-CoNC further indicated the pyrolysis of BZIF into amorphous CoNC (Fig. 1f). The h-CoNC exhibited an ordered porous structure with a bimodal pore size distribution (3.5 and 11.5 nm diameter pores) and Brunner-Emmet-Teller surface area of 293.5 m$^2$ g$^{-1}$ (Fig. 1g and Supplementary Fig. 3). The micropore and mesopore volumes were 0.11 and 0.27 cm$^3$ g$^{-1}$, respectively (Supplementary Table 2); these data illustrate that rich mesopores were present in the thin shells of h-CoNC and that a macro/meso/microporous structure was formed after the chemical-etching process.

**Fine structure of h-CoNC.** XAFS measurements were performed to analyze the coordination environment of Co at an atomic level. The Co K-edge X-ray absorption near-edge structure (XANES)

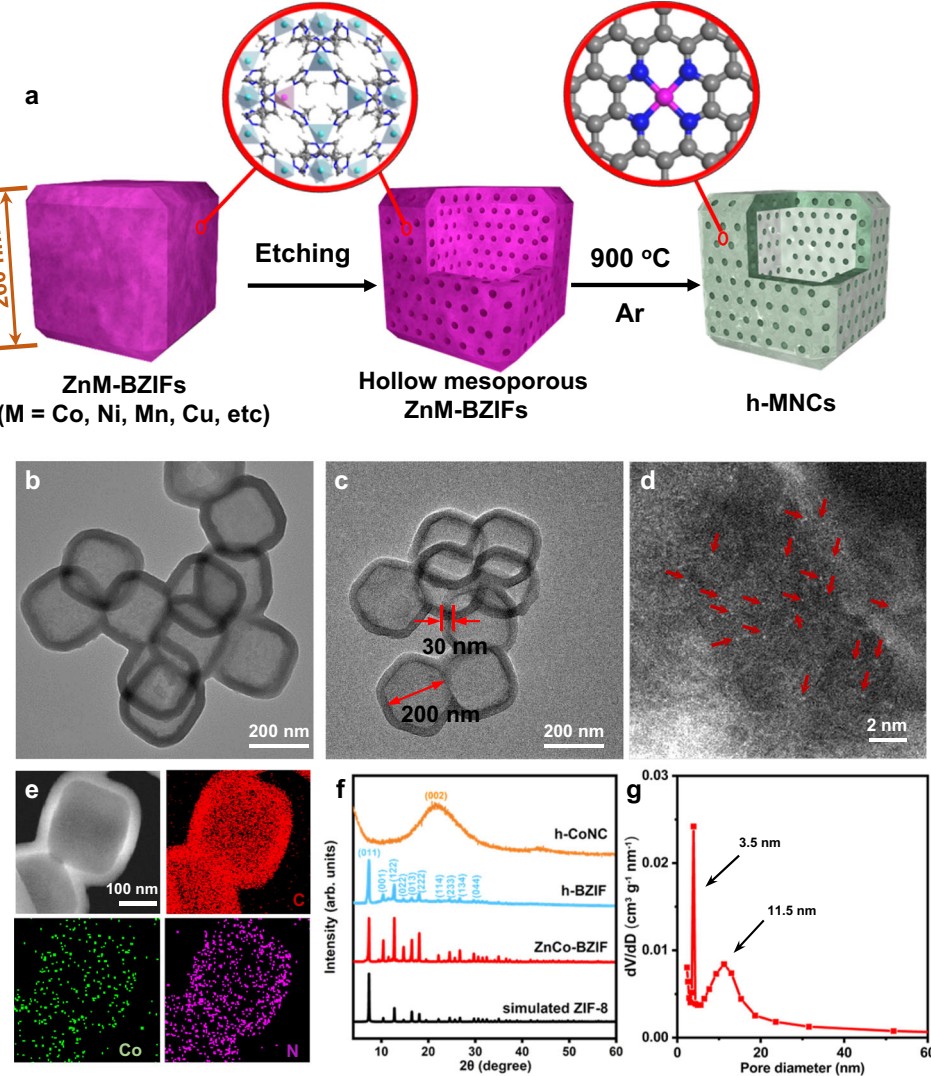

**Fig. 1 Synthesis and characterization of h-CoNC. a** Synthetic scheme for h-MNCs (C, N, metal atoms are shown in gray, blue, and pink, respectively). TEM images of h-BZIF (**b**) and h-CoNC (**c**). **d** AC-HAADF-STEM image of h-CoNC (numerous Co atoms are indicated by red arrows to facilitate identification). **e** EDX mapping of h-CoNC, C (red), Co (green), and N (purple). **f** XRD patterns of h-CoNC, h-BZIF, and ZnCo-BZIF. **g** Pore size distribution of h-CoNC.

spectra of h-CoNC as well as the references including Co foil, CoO, and $Co_3O_4$, are presented in Fig. 2a. The absorption edge of h-CoNC was close to CoO and lower than $Co_3O_4$, indicating that Co atoms were positively charged and that the valence of the Co atoms was ~+2. The Fourier transform (FT) $k^3$-weighted extended XAFS (EXAFS) spectrum of h-CoNC showed one main peak at 1.41 Å, which corresponded to the first coordination shell of Co-N (Fig. 2b). There was no obvious Co-Co peak (2.19 Å) or other high-shell peaks, demonstrating that the Co atoms were atomically dispersed on the nitrogen-doped carbon support. Wavelet transform (WT) was used to analyze Co K-edge EXAFS oscillations. In Fig. 2c, the WT maximum at 4 Å$^{-1}$ for h-CoNC was assigned to Co-N bonding, and with regard to the WT plots of $Co_3O_4$, CoO and Co foil in Fig. 2d–f, no intensity maximum corresponding to Co-Co was detected, further confirming the atomic dispersion of cobalt atoms. Furthermore, EXAFS fittings were carried out to obtain the structural parameters and extract the quantitative chemical configuration of Co atoms in h-CoNC (Supplementary Table 3). The corresponding EXAFS fitting curves of h-CoNC in the R space and k space are shown in Fig. 2g, h, respectively, while the best-fit results of Co foil are displayed in Supplementary Fig. 4. The coordination number of

Co in the h-CoNC was ~4. These fitting results showed that Co atoms were coordinated to four nitrogen atoms in the h-CoNC. In addition, the nature of N in the h-CoNC was studied by X-ray photoelectron spectroscopy (XPS; Supplementary Fig. 5), which indicated that most of the N present in h-CoNC was graphitic N (401.1 eV) with a content of 41.9%, followed by pyridinic N (398.5 eV) with the content of 34.9% and pyrrolic N (399.6 eV) with a content of 23.2%[38]. Based on the EXAFS fitting results and XPS analysis, the chemical structure of the active site of Co-N₄ was proposed as a single Co atom coordinated with four pyridinic N atoms, and the corresponding model is shown in the inset of Fig. 2g.

**The versatility of the approach for various h-MNCs**. To verify the generality of the topo-conversion strategy for constructing hollow mesoporous SACs, we extend this method to other metals, such as Ni, Mn, and Cu. The prepared h-MnNC, h-NiNC, and h-CuNC showed hollow truncated rhombic dodecahedral morphologies similar to those of h-CoNC (Fig. 3a–c), and their XRD patterns only exhibited diffraction peaks generated from the graphitic carbon (Supplementary Fig. 6). As revealed by AC-

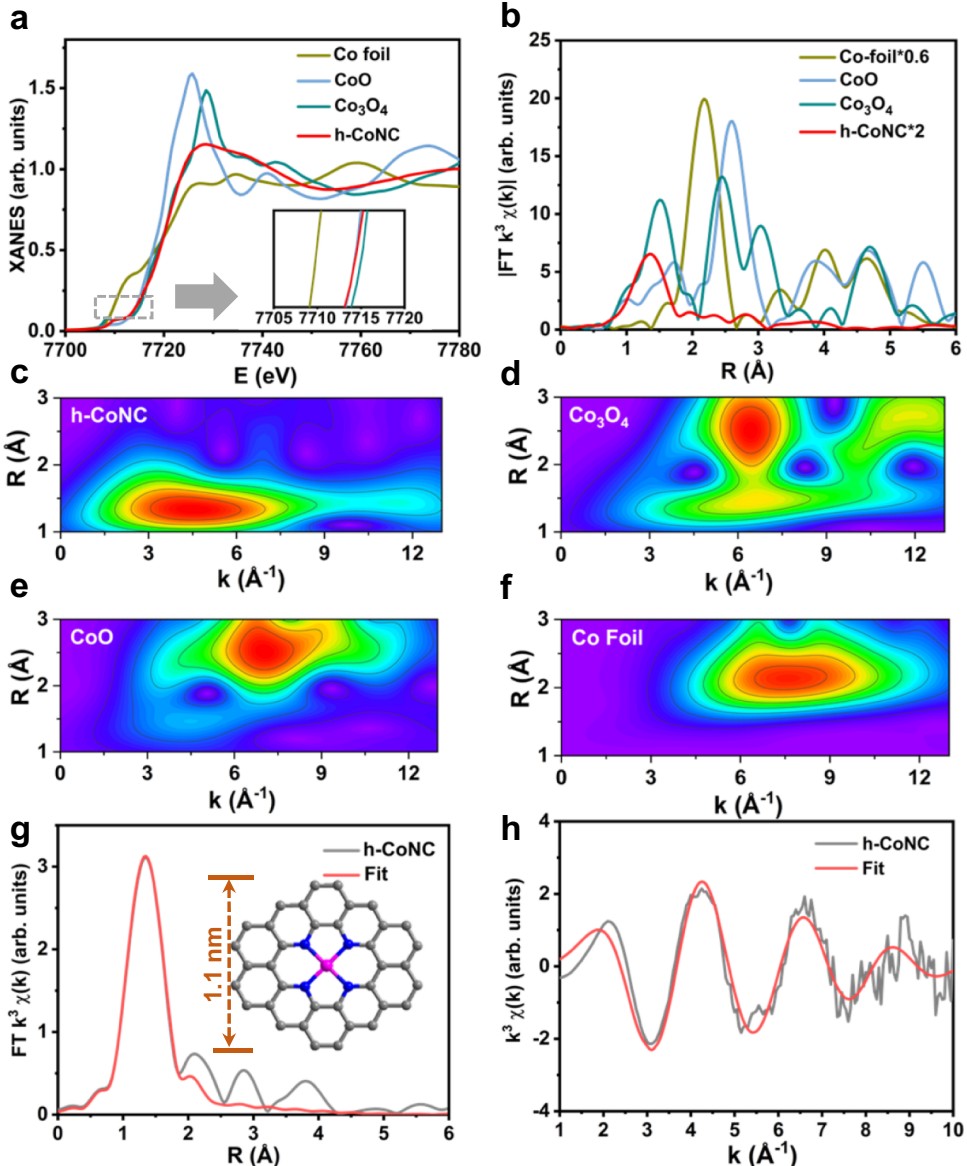

**Fig. 2 XAFS characterization of h-CoNC. a** XANES spectra and **b** Fourier transform at the Co K-edge of h-CoNC, CoO, Co$_3$O$_4$, and Co foil. Wavelet transform of h-CoNC (**c**), Co$_3$O$_4$ (**d**), CoO (**e**), and Co foil (**f**). Corresponding EXAFS fitting curves of h-CoNC in k (**g**) and R (**h**) space. Inset of **g**: Schematic model of h-CoNC: Co (pink), N (blue), and C (gray).

HAADF-STEM, only individual bright dots were observed on the support in h-NiNC, h-MnNC, and h-CuNC (Fig. 3d–f). XAFS results (Fig. 3g–i and Supplementary Fig. 7) further confirmed the absence of metal-metal bonds in all three samples of h-MNC. The metal contents in various h-MNCs measured by ICP–OES are shown in Supplementary Table 1. These results demonstrated the versatility of the approach for various hollow mesoporous atomically dispersed MNC catalysts.

**Catalytic performance of the aniline coupling reaction**. The successful fabrication of hollow mesoporous structured SACs provides an opportunity to catalyze reactions involving large molecular reactants and products highly efficiently. As a proof-of-concept application, oxidative coupling of anilines to azo aromatic compounds was used as the probe reaction (Fig. 4a)[39–41]. Hollow mesoporous h-CoNC was tested compared with the counterparts of hollow NC (h-NC), hollow NC with Co nanoparticles (h-Co$_{NP}$NC), and solid CoNC (s-CoNC), which were derived from hollow ZIF-8, hollow ZnCo-BZIF with Zn/Co ratio

of 20:1, and solid ZnCo-BZIF with Zn/Co ratio of 60:1, respectively (see SI Methods and Supplementary Figs. 8, 9 and 10 for details). As shown in Fig. 4b, h-CoNC exhibited the best performance with 94.4% conversion of aniline within 24 h, followed by h-Co$_{NP}$NC (with a conversion of 88.1%) and s-CoNC (with a conversion of 86.9%). Only azobenzene was observed as the product for all three samples, demonstrating their high selectivity for the oxidative coupling product (Supplementary Fig. 11). In contrast, the amount of azobenzene formed in the presence of h-NC was nearly negligible, indicating that cobalt is the active site for this aniline coupling reaction. Hence, the turnover frequency (TOF) of aniline molecules on each cobalt atom was calculated (Fig. 4c) to compare the cobalt sites on the three cobalt samples. Interestingly, h-CoNC showed the highest TOF (586 h$^{-1}$) among the three samples, while h-Co$_{NP}$NC showed the lowest TOF (126 h$^{-1}$) because it had the largest metal loading, testifying to the high utilization efficiency of active metal atoms in the atomically dispersed catalyst. Compared with reported materials (Supplementary Table 4), h-CoNC also showed obvious

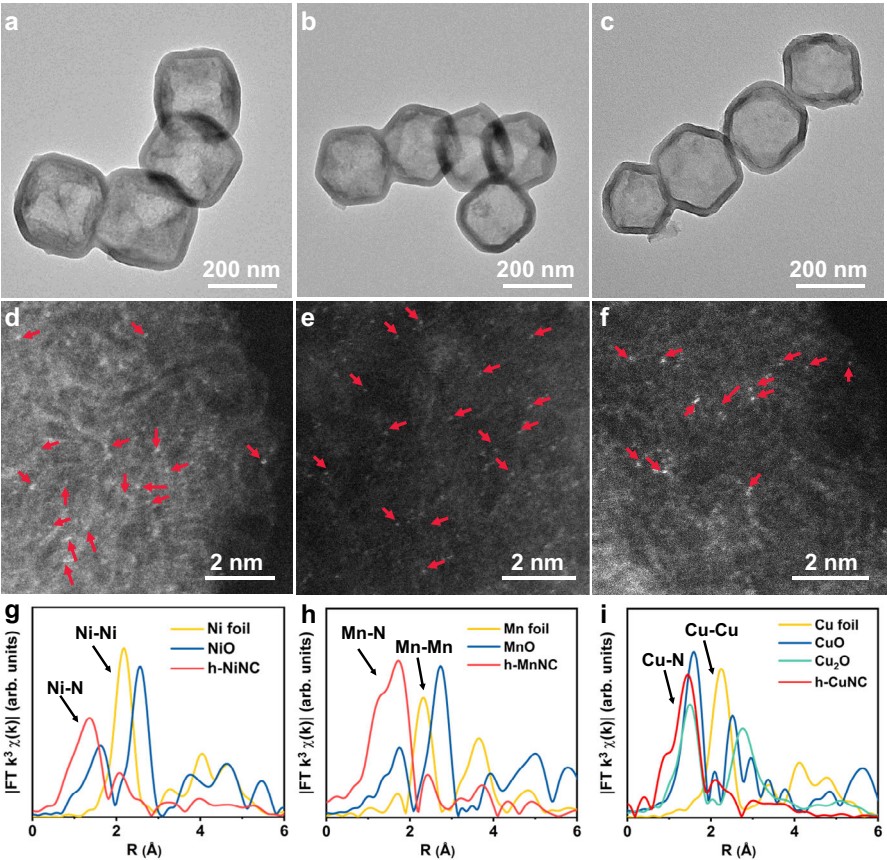

**Fig. 3 Characterization of h-MnNC, h-NiNC, and h-CuNC.** TEM (**a–c**) and AC-HAADF-STEM (**d–f**) images for h-NiNC (**a**, **d**), h-MnNC (**b**, **e**), and h-CuNC (**c**, **f**), respectively (numerous metal atoms are indicated by red arrows to facilitate identification). Fourier transform of the EXAFS spectra: **g** the Ni K-edge of h-NiNC, **h** the Mn K-edge of h-MnNC, **i** the Cu K-edge of h-CuNC.

advantages, as it exhibited the highest TOF under mild operation conditions (room temperature and atmospheric pressure). In addition, the reusability of h-CoNC was also evaluated. Notably, negligible attenuation of catalytic activity was observed with the conversions of aniline remaining as high as 93.7% after five cycles (Fig. 4d). No obvious change in morphology or phase was detected in the TEM images, AC-HAADF-STEM images, or XRD pattern of the reused catalysts (Supplementary Fig. 12). This result proved that our h-CoNC also demonstrated excellent stability without exhibiting aggregation or leaching of the cobalt active sites during the oxidative coupling reaction.

**Computational calculations**. The above catalytic analysis pointed to the critical role of both the atomically dispersed cobalt species and the hollow mesoporous architecture in improving the catalytic performance of h-CoNC; therefore, we investigated these two factors in depth. Computational calculations were first carried out by using density functional theory (DFT) to compare the enhanced catalytic performance of atomically dispersed cobalt sites to their nanoparticle counterparts for aniline oxidative coupling. Models of Co-N$_4$ and Co clusters on graphene were constructed to represent the atomically dispersed cobalt sites (CoNC) and cobalt nanoparticles (Co$_{NP}$NC), respectively (Supplementary Fig. 13). The reaction path (Supplementary Fig. 14) for aniline coupling to azobenzene was calculated[42–44], and the corresponding energies for each step are shown in Fig. 4e. Aniline first adsorbs onto CoNC or Co$_{NP}$NC, and the adsorption is exothermic by 0.01 eV or 1.05 eV, respectively. Next, the adsorbed aniline is activated with a hydrogen atom removed to form an anilino group (i to ii). Then, the nitrogen atom bonds with

another aniline molecule (ii to iii), and the formation of this bond is exothermic by 0.07 and 0.61 eV for CoNC and Co$_{NP}$NC, respectively. The fourth step is the breakage of the N–H bond of the second aniline molecule to form 1,2-diphenylhydrazine (iii to iv), which is the rate-limiting step for both CoNC and Co$_{NP}$NC. CoNC suffers from a much lower reaction barrier (1.32 eV) than Co$_{NP}$NC (1.71 eV). The dissociation of the third hydrogen (iv to v) and the fourth hydrogen (v to vi) are endothermic processes with slightly lower barriers. Finally, the formed azobenzene molecule was desorbed from the catalyst to form the final product (vi to vii), which is easier for CoNC than Co$_{NP}$NC. These computational calculation results well demonstrated that CoNC is kinetically more favorable in the aniline coupling reaction.

**Diffusion behavior and its influence on catalytic performance**. To deeply reveal the structural superiority of the hollow mesoporous structures of h-CoNC, the molecular diffusion behavior was studied. As described above, h-CoNC shows bimodal pore size distribution with pore sizes centered at 3.5 and 11.5 nm, while s-CoNC gained only one pore distribution peak centered at 3.5 nm (Supplementary Fig. 15), which proved the construction of greatly enlarged mesopores in h-CoNC. More importantly, h-CoNC possesses a smaller micropore volume and slightly larger mesopore volume than s-CoNC (Supplementary Table 2), which benefits molecular diffusion since micropores are mostly inaccessible during the coupling reaction. 4-Chloroaniline, an aniline derivative with a Cl substituent that can be easily detected by EDX, was used as a probe in molecular diffusion experiments. Specifically, the h-CoNC and s-CoNC were immersed in the same 4-chloroaniline solution for different time intervals, and the

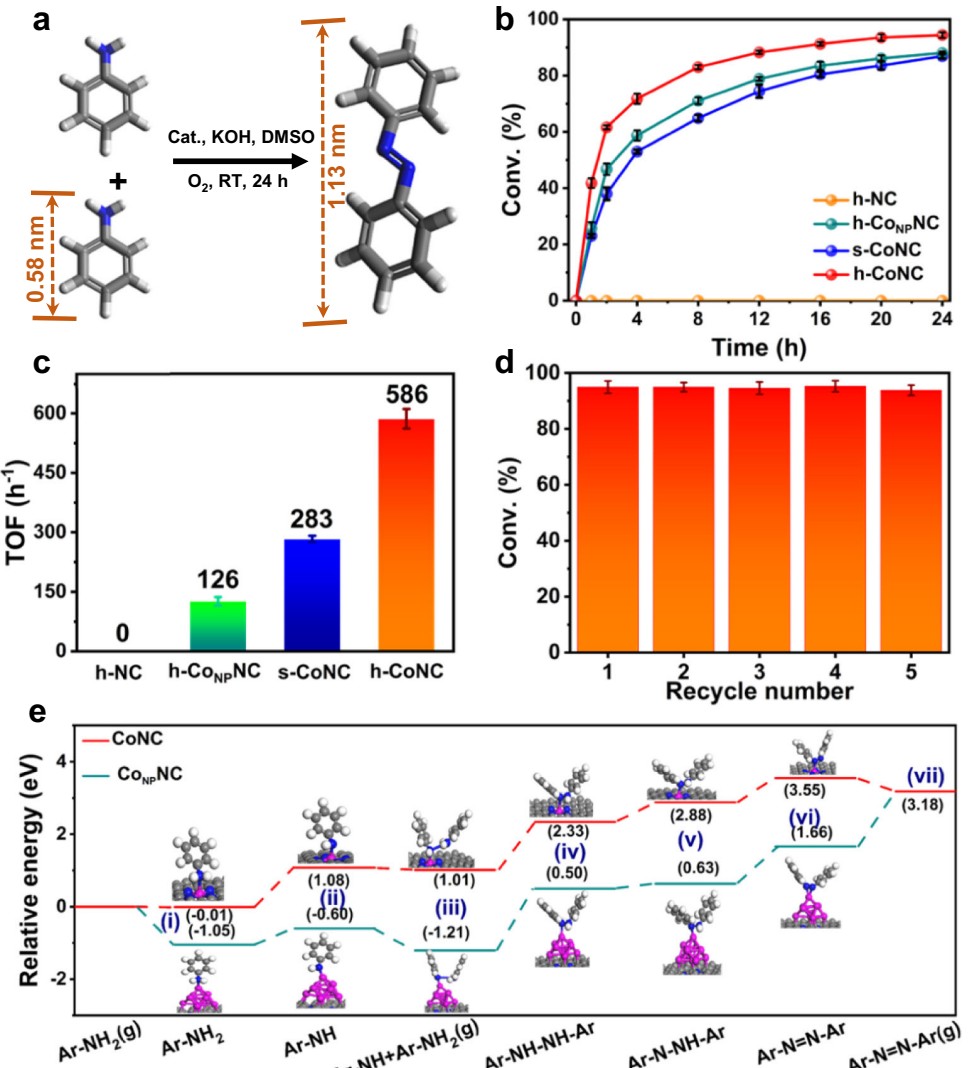

**Fig. 4 Oxidative coupling of anilines. a** Scheme of the catalytic oxidative coupling of anilines to azobenzene. **b** Conversions of aniline coupling using different catalysts over 24 h at 400 rpm. **c** TOF values of different catalysts. **d** Recyclability of h-CoNC. **e** DFT calculations of the reaction path for aniline coupling to azobenzene on Co-NC and Co$_{NP}$NC surfaces (C, N, and Co atoms are shown in gray, blue, and pink, respectively). Error bars represent standard deviations of the means.

relative diffusion rate was estimated by determining the change in the ratio of Cl (from 4-chloroaniline) to Co by EDX. As shown in Fig. 5a, the ratio of Cl: Co increased within 60 min for both h-CoNC and s-CoNC, while h-CoNC always demonstrated much faster diffusion rates (approximately six times higher) than s-CoNC. The reactants were more easily diffused through the hollow mesoporous structures and enriched due to their larger pore size and internal voids.

Considering that diffusion-controlled reactions in solution usually have rather lower apparent activation energies than those limited by chemical reaction kinetics[45], the apparent activation energies of the aniline coupling reaction were calculated from the Arrhenius plots to investigate the apparent reaction kinetics of h-CoNC and s-CoNC, which have the same atomic dispersed active sites. As indicated in Supplementary Fig. 16, h-CoNC exhibited a larger apparent activation energy (39.1 kJ/mol) than s-CoNC (29.8 kJ/mol), corresponding to easier internal diffusion (smaller internal diffusion blockage) in h-CoNC (the external diffusion effect can be excluded at stirring rates larger than 400 revolutions per minute (rpm)) (Supplementary Fig. 17)[46]. The Thiele modulus $\Phi$ and effectiveness factor $\eta$, as indicators of

internal diffusion, were then calculated to quantitively evaluate their internal diffusion (calculated from Eqs. (1), (2), and (3), see the "Methods" section for details). The $\Phi$ and $\eta$ of s-CoNC were 1.11 and 0.63 (Fig. 5b), respectively, indicating that s-CoNC was in a transition condition where the reaction was greatly influenced by internal diffusion. This diffusion limit prevented the substrate molecules from moving to the interior of the catalyst particles, which causes the dead zone that cannot be utilized in the catalytic process. The etching process in the synthesis of our h-CoNC purposefully removed the dead zones and enlarged the mesopores of the shell, thus leading to improved internal diffusion and utilization (schemed in Fig. 5c), in accordance with the improved $\eta$ of h-CoNC (with a value of 0.99, Fig. 5b). The high value of $\eta$ for h-CoNC brings it into the reaction limited condition, effectively eliminating the influence of internal diffusion on the catalytic performance.

The preferential diffusion in the catalytic aniline coupling of h-CoNC derived from the enlarged mesopores was further studied using aniline derivatives with tunable molecular sizes (Fig. 5d) as substrates. In the case of 4-chloroaniline (with a molecular size of 0.65 nm larger than aniline of 0.58 nm),

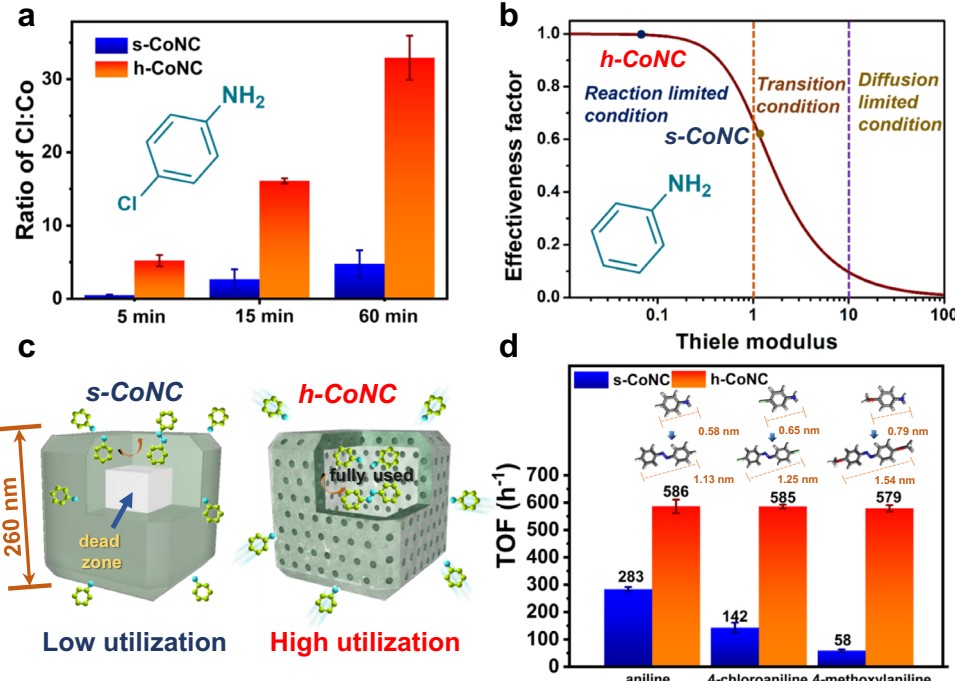

**Fig. 5 Thiele modulus calculation and oxidative coupling of 4-chloroaniline and 4-methoxyaniline. a** Atomic ratio of Cl: Co for h-CoNC and s-CoNC during 4-chloroaniline diffusion at different time intervals. **b** Thiele modulus and effectiveness factor of h-CoNC and s-CoNC. **c** Scheme of the increased diffusion from s-CoNC to h-CoNC. **d** Catalytic performance comparison of aniline, 4-chloroaniline, and 4-methoxyaniline coupling on h-CoNC and s-CoNC. Error bars represent standard deviations of the means.

h-CoNC exhibited 92.5% conversion of 4-chloroaniline to 1,2-bis(4-chlorophenyl)diazene at 24 h (the GCMS data of the main products are shown in Supplementary Fig. 18), while s-CoNC only underwent 22.5% conversion (Supplementary Table 5). The TOF difference between h-CoNC and s-CoNC increased from 2.1 times for aniline to 4.1 times for 4-chloroaniline (Fig. 5d). When an even larger molecule, 4-methoxyaniline (0.79 nm), was used as the substrate, the TOF difference between h-CoNC and s-CoNC increased to 9.9 times. In addition, as the selectivity for oxidative coupling products on h-CoNC remained at 80.6% (Supplementary Table 6; the GCMS data of the main products are shown in Supplementary Fig. 19), the selectivity for the oxidative coupling products of s-CoNC decreased greatly to 46.6% as more oxidation products were generated (Supplementary Fig. 19), which may result from the fact that those small oxidation products are more favored than large coupling products due to spatial hindrance. These results illustrated the advantage of hollow mesoporous structures in catalytic addition reactions involving large organic molecules.

In summary, we have developed a topo-conversion method for synthesizing hollow mesoporous atomically dispersed MNC SACs. This strategy is also applicable to the fabrication of various hollow mesoporous MNC SACs, including Co, Cu, Ni, and Mn. It is noteworthy that h-CoNC exhibited superior catalytic activity and stability in the oxidative coupling of aniline and its derivatives to that of its Co nanoparticle counterpart and solid atomically dispersed s-CoNC counterpart. Computational calculations demonstrated that atomically dispersed cobalt sites have the smallest reaction barrier of the cobalt nanoparticles. Diffusion experiments indicated that the hollow mesoporous structures played an important role in enhancing the diffusion and accessibility of the active site of molecules. Benefitting from the superior intrinsic activity of atomically dispersed Co-N₄ sites and the enhanced diffusion from the mesoporous nanoarchitecture, the TOF for h-CoNC at room temperature reached as high as

586 h⁻¹, surpassing those reported in the literature. This work not only offers an outstanding catalyst that can catalyze the oxidative coupling of aniline to azobenzene and its derivatives, but also sheds light on a method of constructing effective catalysts from hollow mesoporous SACs to address the diffusion of large molecules.

## Methods

**Synthesis of ZnCo-BZIF**. In a typical synthesis, Zn(CH₃COO)₂·2H₂O (295.1 mg, 1.35 mmol) and Co(CH₃COO)₂·4H₂O (5.6 mg, 0.0223 mmol) with a molar ratio of 60:1 were dissolved into 5 mL of H₂O to form a clear pink solution, which was subsequently added into 5 mL of H₂O containing 1.1166 g of 2-MIM (13.6 mmol) and 0.91 mg of CTAB under stirring at room temperature. The resulting mixture turned purple after a few seconds and was left undisturbed at room temperature for 2 h. Then, the product was washed three times with water.

**Synthesis of h-BZIF**. The prepared ZnCo-BZIF was etched with 130 mL tannic acid (TA) solution (5 mg mL⁻¹) while stirring for 20 min. h-BZIF was then washed with water and methanol. The product was then dried in an oven at 80 °C for 4 h.

**Synthesis of h-CoNC**. The h-BZIF was placed in a tube furnace under Ar gas flow, heated to 900 °C (5 °C min⁻¹), and maintained at 900 °C for 2 h.

**Synthesis of h-NiNC, h-MnNC, and h-CuNC**. The synthesis procedure of h-NiNC, h-MnNC, and h-CuNC was similar to that of h-CoNC, except that Co(CH₃COO)₂·4H₂O was substituted by Ni(CH₃COO)₂·4H₂O (5.6 mg), Mn(CH₃COO)₂·4H₂O (5.5 mg), and Cu(CH₃COO)₂·H₂O (4.5 mg), respectively.

**Catalytic oxidative coupling of anilines**. In a typical reaction, a mixture of catalyst (10.0 mg) and KOH (1.0 mmol) was added to a 10 mL Schlenk tube equipped with a rubber stopper. The Schlenk tube was vacuumed with a vacuum pump. The mixture of aniline (1.0 mmol) and DMSO (2 mL) was injected into the Schlenk tube and connected to an oxygen balloon. The reaction mixture was vigorously stirred at 15 °C for 24 h. After the reaction, the products were extracted with ethyl acetate. The catalytic oxidative couplings of other aniline derivatives were the same as the catalytic oxidative coupling of aniline except for the use of 4-chloroaniline (1.0 mmol) or 4-methoxyaniline (1.0 mmol) as reactants. The conversions and selectivities of aniline and its derivatives oxidation coupling were determined by gas chromatography (GC) with a Nexis GC-2030 chromatograph manufactured by

Shimadzu Corporation with a barrier discharge ionization detector and an RTX-5 capillary column (J&W, 30 m, 0.25 mm i.d.). The products were identified using gas chromatography–mass spectrometry (GC–MS) analysis subjected to ISQ GC–MS with an electrical capture detector (ECD, Thermo Trace GC Ultra) using a capillary column (TR-5MS, Thermo Scientific; length 30 m, inner diameter 0.25 mm, film 0.25 μm). The TOF was calculated by dividing the number of converted aniline molecules per hour by the number of cobalt atoms as determined from ICP–OES.

**Calculation of the Thiele modulus $\Phi$ and effectiveness factor $\eta$.** The Thiele modulus $\Phi$ reflects the influence degree of internal diffusion and chemical reactions during the reaction. The effectiveness factor $\eta$ is used to evaluate the utilization efficiency of catalysts and can be derived from $\Phi$. $\Phi$ is defined as Eq. (1) and can be calculated from Eq. (2) while $\eta$ can be calculated from Eq. (3).

$$\Phi = \sqrt{\frac{r_{\text{intrinsc}}}{r_{\text{diffusion}}}} = \frac{\sqrt{n+1}}{2} R_p \sqrt{\frac{k C_{\text{as}}^{n-1}}{D_{\text{eff}}}} \quad (1)$$

where $\Phi$ is the Thiele modulus, $r_{\text{intrinsic}}$ is the maximum reaction rate, $r_{\text{diffusion}}$ is the maximum diffusion rate, $n$ is the reaction order, $R_p$ is the diameter of the catalyst pellet, $k$ is the reaction rate constant, $C_{\text{as}}$ is the surface reactant concentration and $D_{\text{eff}}$ is the effective diffusivity inside the catalyst tunnel.

$$\frac{E_{\text{app,a}}}{E_{\text{int,a}}} = \frac{1}{2} + \Phi \frac{1 - \tanh^2 \Phi}{2 \tanh \Phi} \quad (2)$$

where $E_{\text{app,a}}$ is the apparent activation energy and $E_{\text{int,a}}$ is the intrinsic activation energy. The effectiveness factor $\eta$ was used to evaluate pellet utilization which can be calculated by Eq. (3) when the shape of the catalysts is nearly spherical:

$$\eta = \frac{1}{\Phi} \left[ \frac{1}{\tanh 3\Phi} - \frac{1}{3\Phi} \right] \quad (3)$$

A low $\eta$ means that internal diffusion seriously affects the reaction process and there is a large unused area inside the catalysts.

**Diffusion experiments.** The diffusion experiments were carried out in a 20 mL flask at room temperature. Typically, 10 mg catalysts and 1 mmol 4-chloroaniline were added into 2 mL DMSO under stirring in an Ar atmosphere. Samples were taken at a certain time (5, 15, 60 min). The diffusion amount of 4-chloroaniline was determined by measuring the ratio of Cl: Co by EDX.

## Data availability
The data that support the findings of this study are available from the corresponding author upon reasonable request.

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

## Acknowledgements

This work was financially supported by the National Key Research and Development Program of China (2018YFA0702002), National Natural Science Foundation of China (NSFC), and the Fundamental Research Funds for the Central Universities. The study was also supported by the High Performance Computing Platform of Beijing University of Chemical Technology. The authors thank the BL14W1 station in Shanghai Synchrotron Radiation Facility (SSRF) and the BL1W1B station at Beijing Synchrotron Radiation Facility (BSRF) for XAFS measurement.

## Author contributions

J.L. and A.H. supervised the project; X.H. and T.Z. designed the experiments and analyzed the results. X.H. carried out the computational calculations and most of the experiments. X.W. and Y.Q. synthesized the samples. Z.Z. and B.W. conducted the XAFS measurements. Y.L. helped with the computational calculations. X.H., T.Z., A.H., and J.L. prepared the manuscript. All the authors discussed the results and assisted during the manuscript preparation.

## Competing interests

The authors declare no competing interests.

## Additional information

**Publisher's note** Springer Nature remains neutral with regard to jurisdictional claims in published maps and affiliations.

