## [Peer Review File · Nature Communications]

Title: Hollow mesoporous atomically dispersed metal-nitrogen-carbon catalysts with enhanced diffusion for catalysis involving larger moleculesREVIEWER COMMENTS

Reviewer #1 (Remarks to the Author):

The manuscript reported a general approach to synthesize hollow mesoporous metal-nitrogen-carbon single atom catalysts (SACs) that possessed enhanced diffusion for catalysis involving larger molecules. As a proof-of-concept, the hollow cobalt-nitrogen-carbon SACs demonstrated excellent conversion and selectivity in oxidative coupling of aniline and its derivatives, surpassing the solid counterparts and cobalt nanoparticle counterparts. Moreover, DFT calculations, diffusion experiments and kinetic studies were investigated in detail to unveil the reasons of hollow mesoporous cobalt-nitrogen-carbon for the high performance. On the whole, the manuscript is well written and interesting enough to be considered to publish in Nature Communications. However, minor revision is necessary before it can be finally accepted.

1. The full name should be given when the abbreviation first appears in the manuscript. For example, in line 295 page 10, “.....containing 1.1166 g 2-MiM (13.6 mmol).....”, the full name of 2-MiM should be given.

2. How do the authors confirm the products? GC-MS data or NMR spectroscopy of the products are suggested to be given.

3. Although the author chose aniline selective oxidation as the probe reaction for the activity of the as-prepared h-CoNC catalyst, it is not appropriate to use TON to compare its catalytic performance with other reported catalysts under different reaction times. TOF is recommended to compare the performance of these catalysts. And the recently published work for aniline selective oxidation (Angew. Chem. Int. Ed., DOI: 10.1002/anie.202112907) should be added to the Supplementary Table 4.

4. The following related references published recently on oxidative coupling of anilines are suggested to be added. (1) Angew. Chem. Int. Ed., DOI: 10.1002/anie.202112907; (2) Angew. Chem. Int. Ed., 2021, 60, 6382-6385; (3) J. Am. Chem. Soc., 2021, 143, 2938-2943.

Reviewer #2 (Remarks to the Author):

After revising the paper I do not consider the paper should be published in a highly respected journals, like Nature Communication. The reasons are as follows:

1. The work is good, but they have not written in a very bad manner.

2. The authors forgot to acknowledge the father of the Single Atom Catalysis.

3. For the characterization of compounds, nowhere, the authors described the instrumental details of the instruments used!

4. To ensure Single Atom Catalyst formation, one should use High Resolution TEM, instead of TEM. So I think HRTEM data is necessary

5. The authors mentioned that, they have performed DFT calculations, I suggest instead of DFT calculation, it is always better to use computational Calculations.

6. The Computational details along with the mechanistic modelling should be written in well organized manner. Additionally, the reference for VASP software package is missing.

Reviewer #3 (Remarks to the Author):

There are two major problems with this paper. First, the paper needs to be rewritten by someone who knows English and the science of the work. Second, the references are not representative of the literature. The et al.'s in the references should also have complete names of all authors. Some examples of problems with English just in the first part of the paper, as well as problems with Figures are given below. 20 take out the, experiments. 21 structures 22 factors 34 structures 48 take out the. 50 boost diffusion 53 existing 59 pathways 60 via the ... The discussion is very short. There is no conclusion section. Fig. 1a give a scale. Fig. 1f identify all peaks. Fig. 2g inset, give a scale. Fig. 3 define the red arrows in the caption. Fig. 4a scale, 4b,c,d give error bars on all data points. Fig. 5a,d, give error bars. Fig. 5c give a scale.

Responses to Reviewers

To Reviewer #1:

We really appreciate your insightful advice on our manuscript. We have revised the manuscript corresponding to your pertinent comments. Our answers to your questions are as follows.

Comment 1. *The full name should be given when the abbreviation first appears in the manuscript. For example, in line 295 page 10, “.....containing 1.1166 g 2-MiM (13.6 mmol).....”, the full name of 2-MiM should be given.*

Response: Thanks very much for your kind reminder. We have checked carefully about the abbreviations in the whole manuscript and the corresponding full names were given when they first appeared. These changes are as listed below.

“The target metal atoms separated by the organic linkers and Zn atoms in the **zeolite imidazolate frameworks (ZIFs)** atomically dispersed.....” (page 2)

“a mixture of metal acetates and 2-methylimidazole (**2-MIM**) in a cetyltrimethyl ammonium bromide (CTAB) aqueous solution.” (page 3)

“As shown in the **transmission electron microscope (TEM)** image in Supplementary Fig. 1” (page 4)

“**Computational** calculations were first carried out by using **density functional theory (DFT)** to.....” (page 6)

“The external diffusion effect can be excluded at stirring rates larger than 400 **revolutions per minute (rpm)**” (page 8)

Comment 2. *How do the authors confirm the products? GC-MS data or NMR spectroscopy of the products are suggested to be given.*

Response: Thanks very much for your constructive comment. GC-MS was used to confirm the products. According to your suggestion, these results are added as Supplementary Fig. 11, 18 and 19 in the revised Supplementary Information and the discussions are added in the revised manuscript.

“Only azobenzene was observed as the product for all three samples, demonstrating their high selectivity for the oxidative coupling product (Supplementary Fig. 11).” (paragraph 1 page 6)

“In the case of 4-chloroaniline (with a molecular size of 0.65 nm larger than aniline of 0.58 nm), h-CoNC exhibited 92.5% conversion of 4-chloroaniline to 1,2-bis(4-chlorophenyl)diazene at 24 h (the GCMS data of the main products are shown in Supplementary Fig. 18).” (paragraph 2 page 8)

“In addition, as the selectivity for oxidative coupling products on h-CoNC remained at 80.6% (Supplementary Table 6; the GCMS data of the main products are shown in Supplementary Fig. 19), the selectivity for the oxidative coupling products of s-CoNC decreased greatly to 46.6% as more oxidation products were generated (Supplementary Fig.19), which may result from the fact that those small oxidation products are more favored than large coupling products due to spatial hindrance.” (paragraph 2 page 8)

Supplementary Fig. 11 GC spectra of the products of oxidative coupling of aniline using **a** h-CoNC, **b** s-CoNC and **c** h-Co_{NP}NC as catalysts. Only azobenzene (with the retention time of 7.0 min) was observed as the product (The peak at the retention time of 3.4 min was the reactant aniline). **d** Mass spectrum of the product azobenzene.

Supplementary Fig. 18 GC spectra of the products of oxidative coupling of 4-chloroaniline using **a** h-CoNC and **b** s-CoNC as catalysts. Only 1,2-bis(4-chlorophenyl)diazene (with the retention time of 10.4 min) was observed as the product (The peak at the retention time of 4.7 min was the reactant 4-chloroaniline). **c** Mass spectrum of the product 1,2-bis(4-chlorophenyl)diazene.

Supplementary Fig. 19 GC spectra of the products of oxidative coupling of 4-methoxyaniline using **a** h-CoNC and **b** s-CoNC as catalysts, and mass spectra of **c** 4-nitroanisole (retention time of 5.8 min), **d** N-(4-methoxyphenyl)formamide (retention time of 6.7 min) and **e** 4,4'-dimethoxyazobenzene (retention time of 11.6 min). The

peak at the retention time of 4.8 min was corresponding to the reactant 4-methoxyaniline.

Comment 3. *Although the author chose aniline selective oxidation as the probe reaction for the activity of the as-prepared h-CoNC catalyst, it is not appropriate to use TON to compare its catalytic performance with other reported catalysts under different reaction times. TOF is recommended to compare the performance of these catalysts. And the recently published work for aniline selective oxidation (Angew. Chem. Int. Ed., DOI: 10.1002/anie.202112907) should be added to the Supplementary Table 4.*

Response: Thanks very much for your insightful and constructive suggestion, which help us to improve the quality of the manuscript. TOF is calculated to replace the TON to compare the performance of these catalysts in the revised manuscript in Fig 4c, Fig 5d and Supplementary Table 4. The corresponding descriptions of TOF were revised in the manuscript at page 6 and page 8. In addition, the recently reported work (Angew. Chem. Int. Ed., DOI: 10.1002/anie.202112907) was added in Supplementary Table 4.

“Hence, the turnover **frequency (TOF)** of aniline molecules on each cobalt atom was calculated (Fig. 4c) to compare the cobalt sites on the three cobalt samples. Interestingly, h-CoNC showed the highest **TOF (586 h⁻¹)** among the three samples, while h-Co_{NP}NC showed the lowest **TOF (126 h⁻¹)** because it had the largest metal loading, testifying to the high utilization efficiency of active metal atoms in the atomically dispersed catalyst. Compared with reported materials (Supplementary Table 4), h-CoNC also showed obvious advantages, as it exhibited the highest **TOF** under mild operation conditions (room temperature and atmospheric pressure).” (paragraph 1 page 6)

Fig. 4 Oxidative coupling of anilines. (a) Scheme of the catalytic oxidative coupling of anilines to azobenzene. (b) Conversions of aniline coupling using different catalysts over 24 h at 400 rpm. (c) **TOF values** of different catalysts. (d) Recyclability of h-CoNC. (e) DFT calculations of the reaction path for aniline coupling to azobenzene on Co-NC and Co_{NP}NC surfaces. (C, N, Co atoms are shown in gray, blue and pink, respectively.)

“The **TOF** difference between h-CoNC and s-CoNC increased from 2.1 times for aniline to 4.1 times for 4-chloroaniline (Fig. 5d). When an even larger molecule, 4-methoxyaniline (0.79 nm), was used as the substrate, the **TOF** difference between h-CoNC and s-CoNC increased to 9.9 times.” (paragraph 2 page 8)

Fig. 5 Thiele modulus calculation and oxidative coupling of 4-chloroaniline and 4-methoxyaniline. (a) Atomic ratio of Cl: Co for h-CoNC and s-CoNC during 4-chloroaniline diffusion at different time intervals. (b) Thiele modulus and effectiveness factor of h-CoNC and s-CoNC. (c) Scheme of the increased diffusion from s-CoNC to h-CoNC. (d) Catalytic performance comparison of aniline, 4-chloroaniline and 4-methoxyaniline coupling on h-CoNC and s-CoNC.

Supplementary Table 4. Comparison of the catalytic activity of aniline oxidative coupling of h-CoNC with other reported catalysts.

Catalyst	Temp. (°C)	Oxidant	Conv./Sel. (%)	Metal content (%)	TOF (h ⁻¹)	Reference
h-CoNC	RT ^[a]	O ₂	92.8/100	0.42	586	this work
Au/TiO ₂	100	O ₂ (5 bar)	100/90	1.5	39.2*	Science 2008, 322, 1661.
Ag NPs	25	air	51/100	100	6.7*	ACS Catal. 2013, 3, 478.

Ir(dF-CF ₃ -ppy) ₂ (dtbpy) ⁺	24	air	55/100	34.3	2.8*	J. Am. Chem. Soc., 2021, 143, 2938-2943.
RuO ₂ /Cu ₂ O NPs	RT	air	94/-	1.78	6.7*	ACS Sustainable Chem. Eng. 2018, 6, 11345-11352.
CuBr	60	O ₂	96/-	44.3	5.1*	Angew. Chem. Int. Ed. 2010, 49, 6174.
Meso Mn ₂ O ₃	110	air	99/93	67.0	0.39*	Angew. Chem. Int. Ed. 2016, 128, 2211-2215.
MnOOH	25	O ₂	88/100	62.5	1.1*	J. Mater. Chem. A., 2021, 9, 19692-19697.
BiVO ₄ /g-C ₃ N ₄	27	O ₂	20/82.3	31.7	0.14*	ACS Sustainable Chem. Eng., 2017, 5, 2562-2577.
[N(C ₄ H ₉) ₄] ₂ [Mo ₆ O ₁₉]	60	H ₂ O ₂	99/-	42.2	0.01*	Angew. Chem. Int. Ed. 2021, 60, 6382-6385.
Zr(OH) ₄	100	O ₂	95/94	57.3	2.4*	Angew. Chem. Int. Ed. 2022, 134, e202112907.

^[a]RT= room temperature

Comment 4. The following related references published recently on oxidative coupling of anilines are suggested to be added. (1) *Angew. Chem. Int. Ed.*, DOI: 10.1002/anie.202112907; (2) *Angew. Chem. Int. Ed.*, 2021, 60, 6382-6385; (3) *J. Am. Chem. Soc.*, 2021, 143, 2938-2943.

Response: Thanks very much for your kind suggestion. Recent published related works on oxidative coupling of aniline including these works have been added as reference 39-41 in manuscript.

39. Sitter, J. & Vannucci, K. Photocatalytic oxidative coupling of arylamines for the synthesis of azoaromatics and the role of O₂ in the mechanism. *J. Am. Chem. Soc.* **143**, 2938-2943 (2021).
40. Han, S. et al. Selective oxidation of anilines to azobenzenes and azoxybenzenes by a molecular Mo oxide catalyst. *Angew. Chem. Int. Ed.* **60**, 6382-6385 (2021).
41. Qin, J. et al. Zr(OH)₄-catalyzed controllable selective oxidation of anilines to azoxybenzenes, azobenzenes and nitrosobenzenes. *Angew. Chem. Int. Ed.* **134**, e202112907 (2022).

To Reviewer #2:

We really appreciate your suggestion on our manuscript. We have revised the manuscript corresponding to your pertinent advice.

Comment 1. 1. The work is good, but they have not written in a very bad manner.

Response: Thanks very much for your appreciation and criticism. To improve the quality of the manuscript, scientific polishing and proof-reading are carried out by highly qualified native English speaking editors at AJE recommend by the nature publishing groups (it can be verified on the AJE website using the verification code 799D-0F9F-1191-07BA-303B). The corresponding changes were highlighted in yellow in the manuscript.

Comment 2. The authors forgot to acknowledge the father of the Single Atom Catalysis.

Response: Thanks very much for your kind comment. We acknowledged Zhang, Li and Liu's work (*Single-atom catalysis of CO oxidation using Pt₁/FeO_x. Nat. Chem.* **3**, 634 (2011)) which developed the concept of single-atom catalysis first and add it in the Introduction of the manuscript.

“Since the concept of single-atom catalysts (SACs) was developed by Zhang, Li and Liu et al.,¹ SACs have received tremendous attention in the past decade due to their extraordinary catalytic activity and exclusive selectivity resulting from atomically dispersed metal sites²⁻⁵.”

1. Qiao, B. et al. Single-atom catalysis of CO oxidation using Pt₁/FeO_x. *Nat. Chem.* **3**, 634 (2011).

Comment 3. For the characterization of compounds, nowhere, the authors described the instrumental details of the instruments used!

Response: Thanks very much for your kind comment, which help us to improve the quality of the manuscript. The details of instrument used to characterize the products are now added in the **Methods** section of the revised manuscript. GCMS spectra and the corresponding discussions are added in the revised manuscript.

“The conversions and selectivities of aniline and its derivatives oxidation coupling were determined by gas chromatography (GC) with a Nexis GC-2030 chromatograph manufactured by Shimadzu Corporation with a barrier discharge ionization detector and an RTX-5 capillary column (J&W, 30 m, 0.25 mm i.d.). The products were identified using gas chromatography–mass spectrometry (GC–MS) analysis subjected to ISQ GC–MS with an electrical capture detector (ECD, Thermo Trace GC Ultra) using a capillary column (TR-5MS, Thermo Scientific; length 30 m, inner diameter 0.25 mm, film 0.25 μm).” (paragraph 5 page 10)

“Only azobenzene was observed as the product for all three samples, demonstrating their high selectivity for the oxidative coupling product (Supplementary Fig. 11).” (paragraph 1 page 6)

“In the case of 4-chloroaniline (with a molecular size of 0.65 nm larger than aniline of 0.58 nm), h-CoNC exhibited 92.5% conversion of 4-chloroaniline to 1,2-bis(4-chlorophenyl)diazene at 24 h (the GCMS data of the main products are shown in Supplementary Fig. 18).” (paragraph 2 page 8)

“In addition, as the selectivity for oxidative coupling products on h-CoNC remained at 80.6% (Supplementary Table 6; the GCMS data of the main products are shown in

Supplementary Fig. 19), the selectivity for the oxidative coupling products of s-CoNC decreased greatly to 46.6% as more oxidation products were generated (Supplementary Fig.19), which may result from the fact that those small oxidation products are more favored than large coupling products due to spatial hindrance.” (paragraph 2 page 8)

Supplementary Fig. 11 GC spectra of the products of oxidative coupling of aniline using **a** h-CoNC, **b** s-CoNC and **c** h-Co_{NP}NC as catalysts. Only azobenzene (with the retention time of 7.0 min) was observed as the product (The peak at the retention time of 3.4 min was the reactant aniline). **d** Mass spectrum of the product azobenzene.

Supplementary Fig. 18 GC spectra of the products of oxidative coupling of 4-chloroaniline using **a** h-CoNC and **b** s-CoNC as catalysts. Only 1,2-bis(4-chlorophenyl)diazene (with the retention time of 10.4 min) was observed as the product (The peak at the retention time of 4.7 min was the reactant 4-chloroaniline). **c** Mass spectrum of the product 1,2-bis(4-chlorophenyl)diazene.

Supplementary Fig. 19 GC spectra of the products of oxidative coupling of 4-methoxyaniline using **a** h-CoNC and **b** s-CoNC as catalysts, and mass spectra of **c** 4-nitroanisole (retention time of 5.8 min), **d** N-(4-methoxyphenyl)formamide (retention

time of 6.7 min) and **e** 4,4'-dimethoxyazobenzene (retention time of 11.6 min). The peak at the retention time of 4.8 min was corresponding to the reactant 4-methoxyaniline.

Comment 4. To ensure Single Atom Catalyst formation, one should use High Resolution TEM, instead of TEM. So I think HRTEM data is necessary

Response: Thanks very much for your kind comment. To confirm the formation of the single atom site of s-CoNC and reused h-CoNC respectively, the AC-HAADF-STEM images have been supplemented in the Supplementary Information.

Supplementary Fig. 10 Characterization of s-CoNC. **a** TEM image, **b** AC-HAADF-STEM image (numerous Co atoms are indicated by red arrows to facilitate identification) and **c** XRD pattern of s-CoNC. Only discrete bright dots were observed on the AC-HAADF-STEM image, demonstrating the formation of atomic dispersion of Co atoms of s-CoNC. XRD pattern of h-CoNC demonstrated the absence of metallic cobalt in s-CoNC.

Supplementary Fig. 12 Characterization of reused h-CoNC. **a** TEM image, **b** AC-HAADF-STEM image (numerous Co atoms are indicated by red arrows to facilitate identification) and **c** XRD pattern of the h-CoNC after five runs of coupling reactions. No obvious change was observed from the TEM image and XRD pattern of the reused

catalysts and the AC-HAADF-STEM image shows atomic dispersion of Co atoms, indicating the excellent stability of h-CoNC.

Comment 5. The authors mentioned that, they have performed DFT calculations, I suggest instead of DFT calculation, it is always better to use computational Calculations.

Response: Thanks very much for your kind suggestion. According to your suggestion, we used “Computational calculations” instead of “DFT calculations” to describe the theoretical results.

Comment 6. The Computational details along with the mechanistic modelling should be written in well organized manner. Additionally, the reference for VASP software package is missing.

Response: Thanks very much for your kind comments. More computational details about the mechanistic modelling are added in the experimental section and the references of the VASP software package are added as references 1, 2 and 7 in the revised Supplementary Information. These changes are listed below.

“Computation methods: All the spin-polarized calculations were performed using the Vienna ab initio Simulation Package (VASP)^{1, 2} with projector-augmented waves (PAW) pseudopotentials and the exchange-correlation functionals parametrized by Perdew, Burke, and Ernzerhof for the generalized gradient approximation (GGA)³⁻⁵. Dispersion - corrected density functional theory calculations (DFT-D3)⁶ were performed for group adsorbed on Co-N₄/G and Co cluster) surfaces. We set an energy cutoff, a convergence criterion for self-consistent iteration and ionic relaxation to be 480 eV, 10⁻⁵ eV and 0.05 eV Å⁻¹, respectively. The k-space integration was performed using a 3 × 3 × 1 Monkhorst-Pack grid. For the catalytic reaction, the model of Co-NC is built by coordinating Co atoms with four nitrogen atoms to form a square planar Co-N₄ structure on nitrogen-doped graphene, as determined from the fitting results of EXAFS data. The cluster model with ten Co atoms placed on nitrogen-doped graphene was applied to represent the Co_{NP}NC. The entropy contributions of small molecules (TAS), including the vibrational, rotational and translational entropies, were considered

to estimate the Gibbs free energy change (ΔG) at the temperature of 298 K. The ΔG values of elementary steps were estimated according to $\Delta G = \Delta H - T\Delta S$. The reaction enthalpy (ΔH) was approximated with the total energy difference (ΔE), neglecting the small zero-point energy correction (ΔZPE), heat capacity correction and $\Delta(pV)$ term.⁷ (Supplementary Information)

1. Kresse G & Furthmüller J. Efficiency of ab-initio total energy calculations for metals and semiconductors using a plane-wave basis set. *Comput. Mater. Sci.* **6**, 15-50 (1996).
2. Kresse G & Joubert D. From ultrasoft pseudopotentials to the projector augmented-wave method. *Phys. Rev. B* **59**, 1758-177 (1999).
7. Jin J. et al. Insight into room-temperature catalytic oxidation of nitric oxide by Cr_2O_3 : A DFT study. *ACS Catal.* **8**, 5415-5424 (2018).

To Reviewer #3:

We really appreciate for your advice on our manuscript. We have revised the manuscript corresponding to your pertinent comments. Our answers to your questions are as follows.

Comment 1. There are two major problems with this paper. First, the paper needs to be rewritten by someone who knows English and the science of the work.

Response: Thanks very much for your comments. According to your suggestion, scientific polishing and proof-reading are carried out by highly qualified native English speaking editors at AJE recommend by the nature publishing groups (it can be verified on the AJE website using the verification code 799D-0F9F-1191-07BA-303B). The corresponding changes were highlighted in yellow in the manuscript.

Comment 2. Second, the references are not representative of the literature. The et al.'s in the references should also have complete names of all authors.

Response: Thanks very much for your comment. According to the requirements of the Nature Communication journal, “all authors should be included in references lists unless there are six or more, in which case only the first should be given, followed by

'et al.'. Authors should be listed last name first, followed by a comma and initials (followed by full stops) of given names". Hence, we revised the references accordingly and the representative literature were added in references.

Comment 3. *Some examples of problems with English just in the first part of the paper, as well as problems with Figures are given below. 20 take out the, experiments. 21 structures 22 factors 34 structures 48 take out the. 50 boost diffusion 53 existing 59 pathways 60 via the ...*

Response: Thanks very much for your advice, which helps us to improve the quality of the manuscript.

The English problems are corrected and marked red in the manuscript. These changes are listed below.

- "the diffusion and kinetic experiments" (line 20)
- "hollow mesoporous structures" (line 21)
- "metal-nitrogen-carbon (MNC) structures" (line 34)
- "the narrow micro/mesoporous structures" (line 44)
- "~~boosting~~ the diffusion" (line 50)
- "the existing mesopores" (line 53)
- "short diffusion pathways" (line 58)
- "via the hard template method" (line 59)
- "hierarchically porous structures" (line 77)
- "hierarchically porous structures" (line 82)

Comment 4. *The discussion is very short. There is no conclusion section.*

Response: We sincerely appreciate your constructive suggestion. We add the section of **Results and discussion** to make a thorough discussion of our results. The section of **Conclusion** was also added behind the **Results and discussion** to make a comprehensive conclusion of the whole article.

Comment 5. *Fig. 1a give a scale. Fig. 1f identify all peaks. Fig. 2g inset, give a scale.*

Fig. 3 define the red arrows in the caption. Fig. 4a scale, 4b, c, d give error bars on all data points. Fig. 5a,d, give error bars. Fig. 5c give a scale.

Response: We sincerely appreciate your constructive suggestion. All these Figures have been revised according to your suggestion. The revised figures and captions are listed below.

Fig. 1 Synthesis and characterization of h-CoNC. (a) Synthetic scheme for h-MNCs (C, N, Co atoms are shown in gray, blue and pink, respectively). TEM images of h-BZIF (b) and h-CoNC (c). (d) AC-HAADF-STEM image of h-CoNC (numerous Co atoms are indicated by red arrows to facilitate identification). (e) EDX mapping of h-

CoNC, C (red), Co (green), and N (purple). (f) XRD patterns of h-CoNC, h-BZIF, and ZnCo-BZIF. (g) Pore size distribution of h-CoNC.

Fig. 2 XAFS characterization of h-CoNC. (a) XANES spectra and (b) Fourier transform at the Co K-edge of h-CoNC, CoO, Co₃O₄, and Co foil. Wavelet transform of h-CoNC (c), Co₃O₄ (d), CoO (e) and Co foil (f), respectively. Corresponding EXAFS fitting curves of h-CoNC in k (g) and R (h) space. Inset of (g): Schematic model of h-CoNC: Co (pink), N (blue), and C (gray).

Fig. 3 Characterization of h-MnNC, h-NiNC and h-CuNC. TEM (a-c) and AC-HAADF-STEM (d-f) images for h-NiNC (a, d), h-MnNC (b, e) and h-CuNC (c, f), respectively (numerous metal atoms are indicated by red arrows to facilitate identification). Fourier transform of the EXAFS spectra: (g) the Ni K-edge of h-NiNC, (h) the Mn K-edge of h-MnNC, (i) the Cu K-edge of h-CuNC.

Fig. 4 Oxidative coupling of anilines. (a) Scheme of the catalytic oxidative coupling of anilines to azobenzene. (b) Conversions of aniline coupling using different catalysts over 24 h at 400 rpm. (c) TOF values of different catalysts. (d) Recyclability of h-CoNC. (e) DFT calculations of the reaction path for aniline coupling to azobenzene on Co-NC and Co_{NP}NC surfaces. (C, N, Co atoms are shown in gray, blue and pink, respectively.)

Fig. 5 Thiele modulus calculation and oxidative coupling of 4-chloroaniline and 4-methoxyaniline. (a) Atomic ratio of Cl: Co for h-CoNC and s-CoNC during 4-chloroaniline diffusion at different time intervals. (b) Thiele modulus and effectiveness factor of h-CoNC and s-CoNC. (c) Scheme of the increased diffusion from s-CoNC to h-CoNC. (d) Catalytic performance comparison of aniline, 4-chloroaniline and 4-methoxyaniline coupling on h-CoNC and s-CoNC.

REVIEWERS' COMMENTS

Reviewer #1 (Remarks to the Author):

The manuscript has been well revised and can be accepted in its present form.

Reviewer #2 (Remarks to the Author):

Comment in response to the revised manuscript

The questions are addressed properly.

The manuscript is now written in well-organized way.

Thus, in conclusion, after checking the manuscript, I suggest that the manuscript can be published in the present form.

Responses to Reviewers

To Reviewer #1:

We really appreciate for your approval on our manuscript.

Comment 1. The manuscript has been well revised and can be accepted in its present form.

Response: Thanks very much for your approval to our works. We are glad that our responses are satisfactory to you. We are really grateful for your questions in the first round, which helps us improve the quality of our manuscript.

To Reviewer #2:

We really appreciate for your approval on our manuscript.

Comment 1. The questions are addressed properly. The manuscript is now written in well-organized way. Thus, in conclusion, after checking the manuscript, I suggest that the manuscript can be published in the present form.

Response: Thanks very much for your approval to our works. We really thanks for your good questions and kind comments in the first round. Our manuscript has been improved a lot with the help of your comments.